# Time to non-adherence to iron and folic acid supplementation and associated factors among pregnant women in Hosanna town, South Ethiopia: Cox-proportional hazard model

Belay Bancha[1]*, Bereket Abrham Lajore[2], Legese Petros[1], Habtamu Hassen[1], Admasu Jemal[3]

1 Lecturer of Human nutrition, Hosanna Health Science College, Hosanna, Ethiopia, 2 Lecturer of Biostatistics, Hosanna Health Science College, Hosanna, Ethiopia, 3 Lecturer of Epidemiology, Hosanna Health Science College, Hosanna, Ethiopia

* banchabelay@gmail.com

**Data Availability Statement:** All relevant data are within the manuscript and its Supporting Information files.

## Abstract

### Backgrounds

Micronutrient deficits in women of reproductive age have been linked to poor pregnancy outcomes. The most common micronutrient deficits in women are iron and folate. The World Health Organization recommends daily oral iron and folic acid supplementation (IFAS) as part of routine antenatal care to lower the risk of maternal anemia and adverse pregnancy outcomes. However, the effectiveness of the supplementation relies on client's strict adherence. The aim of this study was to determine time- to- non-adherence to IFAS and associated factors among pregnant women in Hosanna Town, South Ethiopia.

### Methods

A community based cross sectional study design was employed from May 15-June11, 2021. Data were entered into Epi-Data version 3.1 and exported to SPSS version 23 for analysis. The Cox regression hazard model was applied. The threshold of statistical significance was declared at a p-value <0.05 and adjusted hazard ratios (AHRs) with corresponding 95% confidence intervals were used to report.

### Result

The median time-to-non-adherence was 74 days (95 percent CI: 65.33–82.67). After adjusting for the confounders, age (AHR = 1.05, 95% CI: 1.01–1.09), education status (AHR = 2.43 95%CI 1.34–4.40, AHR 3.00, 95% CI: 2.09–4.31, AHR 1.91, 95% CI: 1.32–2.77), household's wealth index (AHR = 1.73, 95% CI: 1.19–2.51, AHR = 1.64, 95% CI:1.15–2.35), and counseling at service delivery (AHR = 2.53, 95% CI: 1.88–3.41) were independent predictors of time to non-adherence to IFAS among pregnant women.

**Funding:** The author(s) received no specific funding for this work

**Competing interests:** The authors have declared that no competing interests exist

**Abbreviations:** AHR, Adjusted Hazards Ratios; ANC, Antenatal Care; CHR, Crude Hazards Ratios; GA, Gestational Age; IFAS, Iron Folic Acid Supplementations; LNMP, Last Normal Menstrual Period; MN, Micronutrients; PCA, Principal Component Analysis; WHO, World Health Organizations; CI, Confidence Interval.

## Conclusion

The median time to non-adherence was short and women became non-adherent before the recommended duration. Improving women's education and counseling pregnant women on IFAS during pregnancy would make a change.

## Introduction

Micronutrient (MN) deficits in women of reproductive age have been linked to poor pregnancy outcomes and offspring growth and development. The most common MN deficits in women are iron and folate. Iron deficiency is well documented to have negative effects on productivity and cognition in general population, and it is the major cause of anemia during pregnancy, accounting for 20% of all maternal and perinatal death, as well as low birth weight. Folate deficiency during pregnancy can result in neural tube defects in newborns and other adverse pregnancy outcomes. Supplementation of the two MNs is frequently suggested for pregnant mothers since both forms of nutritional deficits can be prevented and treated [1–4].

To lower the risk of low birth weight, maternal anemia, and adverse pregnancy outcomes, the World Health Organization (WHO) highly recommends daily oral iron and folic acid supplementation (IFAS) as part of routine antenatal care. According to the guidelines, all pregnant women in all settings should receive 30–60 mg of elemental iron and 400 μg (0.4 mg) folic acid during pregnancy, starting as soon as feasible as part of standard antenatal care [5–7].

In accordance with WHO's recommendations, the routine antenatal care (ANC) program of the Ethiopian Government suggested daily IFAS during pregnancy as early as feasible for a healthy pregnancy outcome [8]. The effectiveness of IFAS relies on client's strict adherence [9–11], which is defined as when pregnant women attending prenatal clinics used IFA pills for at least 4 days per week prior to the survey date [12] or for > 90 days at third trimester of pregnancy [13].

Despite proved benefits [3, 14] and established international and national guidelines, evidences show that pregnant women are non-adherent to IFAS both in rural and urban settings [14, 15]. Previous research, however, did not show the time to non-adherence (time to event). Therefore, this study was aimed to generate evidence on median time to non-adherence to IFAS and associated factors among pregnant women.

## Materials and methods

### Setting

The research was carried out in Hosanna Town, Southern Nations Nationalities and People Regional State (SNNPR) of Ethiopia. The town is situated in 230 kilometers to the south of national capital, Addis Ababa. According to 2007 census [16], total population of the town was 69,957; 35, 503 were males and 34, 454 were females. In the same census, the population growth rate in the region was 2.9% per year. Based on this, the projected total population of the Town for 2021 was 104,387. In the region, 23.3% and 3.5% population are women in their reproductive age and expected to be pregnant respectively [17]. Based on these evidences, there were 24,323 women of reproductive age (15–49 years) and 3,654 estimated pregnancies for year 2021.

### Study design and population

The research involving community based cross sectional study design was employed between May 15 and June 11, 2021. The source population consisted of all pregnant women in the

research area and all pregnant women in selected sub-cities were sample population. The inclusion criteria considered all pregnant women in the study cluster who were booked for ANC one-week preceding the study. Pregnant women who were booked for ANC but whose registration for follow-up was less than one week prior the survey date were excluded. Also women who unable to recall their last normal menstrual cycle or gestational age (GA) at the time of booking were not allowed participating.

## Sample size determination and sampling technique

The sample size was calculated with Epi Info version 7 using the double population proportion formula to detect a non-adherence rate of at least 25.1%, [14], 95% significance level and 5% margin of error; a sample size of 289 was obtained. The Cox proportional hazards model (power cox) was used to determine sample size for factors associated with non-adherence to IFAS using Stata version 15.0 considering the presence of censoring and adjusting for others. Non- adherence was considered a failure (outcome). Factors obtained from literatures having a significant association with adherence to IFAS were considered for sample size calculation; knowledge about IFAS [6], counseling on IFAS [12], Partner support [18], and Educational status [19]. After computing various factors, sample size calculated for educational status was 139; the largest sample size computed for factors associated with IFAS adherence. Therefore, the minimum sample size required for the non-adherence estimation would be 289. Considering 5% non-response rate [14, 20, 21] and design effect of 1.5, the final sample size was (289* +5%) *1.5 = 456.

$$n = \frac{(z\alpha/2)^2 \times p(1-p)}{d^2} = \frac{\left(\frac{z\alpha}{2}\right)^2 \times 0.251(1 - 0.251)}{(0.05)^2}$$

$$= \frac{(1.96)^2 \times 0.188}{0.0025} = 289$$

Where n is required sample size, z$\alpha$/2 is 95% CI, p is population proportion and d is margin of error.

Study clusters were identified using a two-stage cluster sampling procedure. After randomly selecting five Kebeles (the smallest administrative structure), total number of predefined distinct clusters (Mender) (smallest cluster within an administrative Kebele) were obtained, after which we obtained the size of the pregnant women for each cluster. The required number of clusters from each Kebele were assigned using probability proportional to population size approach, in which larger settlements have a higher chance of being selected as clusters. Reserve clusters were used until we obtain required sample size (Fig 1).

## Data collection tool and procedure

Ethiopian Demographic and Health Survey 2016 (EDHS,2016) [22] and relevant literature [6, 11, 14, 18, 23] were used to adapt data collection tool.

A household's wealth status was computed based on 23 household assets and housing quality variables which were adapted from EDHS2016 [22], given that the study setup is urban. First, all the study participants were asked about the ownership of assets by their respective households. Those who owned the asset received a score of "1," while those who did not received a score of "0". A structured questionnaire was prepared in English, translated to Amharic, and then back translated into English to ensure consistency in order to measure the required parameters. The Amharic version tool was then employed.

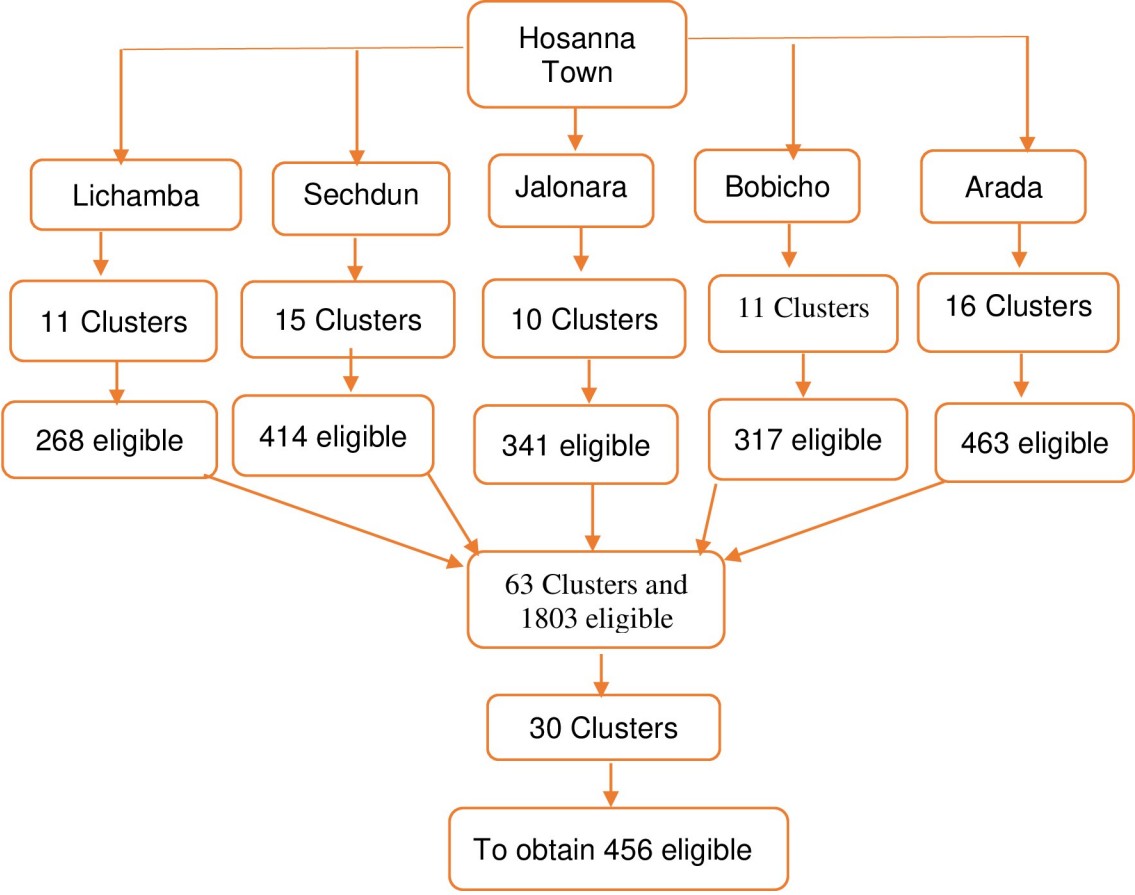

**Fig 1. Diagrammatic representation of sampling technique, Hosanna town, 2021.**

## Operational definitions

**Non-adherence.** When a pregnant woman visiting an antenatal clinic took IFA tablets for less than four days per week for the week prior to the survey or for less than 90 days during the third trimester of pregnancy [12, 13].

**Censored.** Pregnant women who were adherent at the time of data collection were censored observations.

**Event.** When a pregnant lady took IFAS tablet for less than four days one week preceding the survey date, the event occurred.

**Length of stay.** Is the amount of days a pregnant woman contributed while on IFAS until she experienced an event of interest (non-adherent) or censorship.

## Study variables

The dependent variable was time to non- adherence of IFA supplementation measured in days. Censored observations were denoted by 0, whereas events were indicated by 1. The period of time a pregnant woman spent on IFAS in days was determined as the difference between the entire GA (from LNMP to the day of data collection) and the calculated and/or reported GA at initial booking for ANC. Maternal age and educational status, household's wealth index, counseling status of health institution on IFAS, knowledge on IFAS, waiting time to receive care, frequency of ANC visit, history of adverse fetal outcomes *and* history of

anemia were variables hypothesized to be independent predictors of time to non-adherence which is a primary outcome in this study.

## Data quality assurance

To ensure data quality, ten data collectors and one supervisor received two days training on tool clarity and overall data collection processes. The training was emphasized on sociodemographic information's, household's wealth status, obstetric factors, Personal exposure to media, health facility related factors, and Knowledge on IFAS and anemia. Structured questionnaire was prepared in English, translated into Amharic, and then back translated into English to ensure consistency. A pre-test was conducted on 5% of the sample size in a nearby town. Cronbach's alpha was done to assess internal consistency (alpha coefficient for household wealth status (23 items) = 0.85, media access (3 items) = 0.71, counseling at health facility (7 items) = 0.76, Knowledge on IFAS (15 items) = 0.79). A public health officer supervised the data collection process, while the primary investigators (PIs) supervised the whole technique. All collected data were handled to PIs and checked and cleaned for consistency and completeness; daily discussion was held in case of inconsistencies.

## Data processing and analysis

Epi-Data version 3.1 for data entry and Statistical Package for Social Science (SPSS) version 23 for analysis were use. Before analysis missing value, new categories and normality for continuous variables were checked. Households wealth index was computed by principal component analysis (PCA) based on household assets and housing quality variables which adapted from EDHS 2016 [22]. Pregnant women's knowledge on IFAS was computed after performing PCA based on 15 items. Problematic variables were removed step by step, eleven items having four component factors that explains a total variance of 64.3% were retained; whose alpha coefficient was 0.78, all having acceptable correlation matrix (KMO = 0.78, x2 = 1048, P <0.001), sampling adequacy of each item was > 0.5. The value of retained variables was aggregated and used median as a cut off to declare knowledge status of study population.

The difference between total GA (spanning from last normal menstrual cycle to date of data collection) and the calculated and or reported GA at first booking for ANC was taken as total time contributed in days during which a pregnant woman was on IFAS. Survival curve was used to display the survival status (time to non-adherence) among different characteristics.

For survival analysis, the outcome variable was dichotomized to event and censored. The assumptions of proportional hazard were tested statistically and graphically. Against each categorical variable, we performed the log-og survival plot and the Kaplan-Meier survival plot. Both log-log survival plot and Kaplan-Meier survival and predicted plot revealed that the plots were parallel to each other. We have also conducted Schoenfeld test with the corresponding p-value for all variables. The Kaplan-Meier test was used to assess the median survival time between groups. The multivariate Cox Proportional Hazard model was used to examine the factors associated with time to non-adherence. The crude and adjusted hazards ratios with a 95% confidence interval (CI) were used as a measure of effect size. The Cox proportional hazard model assumption was tested graphically using log-minus-log survival plots against time for predictors. Multivariable Cox proportional hazard regression model was used to control the confounding effect of variables. In bivariate analysis, variables having a p-value < 0.25 were selected as potential predictors and used in multivariable analysis. A p-value < 0.05 with a corresponding 95% CI was declared statistically significant.

### Ethics approval and consent to participate

The study was approved by the Institutional Review Board (IRB) of Hosanna Health Science College. In addition, permission was obtained from health department of the local government offices. Informed written consent was obtained from all participants. Respondents were informed that they had the right to refuse or discontinue the interview. The information provided by each respondent was kept confidential. Women who were non-adherent at the time of data collection were successfully counseled on the benefits of IFAS.

## Results

### Socio-demographic characteristics

The study comprised a total of 426 pregnant women, with a 93.4% response rate. The mean (± SD) age of pregnant women was 28.64 (± 4.5). In this study, almost all pregnant women (98.8%) were married, and 405 (95%) have attended formal education, 202 (47.4%), 156 (36.6%), 39 (9.2%) and 29 (6.8%) were housewives, employee, merchant and others respectively. Among the study population, 71.8% of all participants were followers of Protestant Christianity by religion and 70.9% were Hadiya ethnic. Nearly half 193 (45.3%) of household were composed of 5 or more family size. Participants' household wealth index status was ranked; the highest, middle and the lowest tertiles, respectively, were represented by 123 (28.9%), 145 (34%) and 158 (37.1%) wealth index score (Table 1).

### Pregnancy related conditions

In this study, 63 (14.8%) of the current pregnancy was not planned. The study also showed that only 44 (10.3%) of pregnant women booked for ANC in recommended beforehand sixteen weeks of GA. The median GA at first ANC booking was 20 weeks. Public Health institutions were predominant for ANC preference. This study documented that about eight in ten (78.4%) study participants ever skipped iron folic acid supplementation for various reasons. The most common reported reason for skipping was gastric irritation (56.3%) followed by forgetfulness (51.2%). This study revealed that 85 (20%) mothers get pregnant for the first time; so far 340 (79.8%) mothers gave birth to at least one live birth. In the current study, participants reported that 10.6%, 3.5%, 3.3% and 1.9% had history of abortion, still birth, low birth weight and preterm birth respectively. The reported prevalence of anemia in the current pregnancy among participant women accounts for 26.5%.

### Personal characteristics in relation to information access

In this study women's access to information was assessed and less than one in three watch televised media, listen to radio and read medical magazine at least once on a weekly frequency (Table 2).

### Counseling status and client's knowledge on IFAS

Counseling status in health delivery system was assessed using seven items and in half of the cases counseling at service delivery was labeled as poor. The median waiting time to obtain ANC service was found to be 40 minutes. Participants knowledge status on IFAS was labeled as poor and good in 252 (59.2%) and 174 (40.8%) cases respectively.

### Survival analysis

In this study, 426 pregnant women participated; contributed for 23,367 maternal-days of observations. From a total, 226 (53.1%) were non-adherent and the rest 200 (46.9%) were censored.

**Table 1. Socio-demographic characteristics of pregnant women in Hosanna town, 2021 (n = 426).**

| Variables | Category | Frequency | % |
|---|---|---|---|
| Marital status | Married | 421 | 98.8 |
| | Single | 3 | 0.7 |
| | Divorced | 2 | 0.5 |
| Religion | Protestant | 306 | 71.8 |
| | Orthodox | 87 | 20.4 |
| | Muslim | 14 | 3.3 |
| | Catholic | 16 | 3.8 |
| | Others | 3 | 0.7 |
| Ethnicity | Hadiya | 302 | 70.9 |
| | Kembata | 51 | 12 |
| | Siltie | 21 | 4.9 |
| | Amahara | 33 | 7.7 |
| | Others | 19 | 4.5 |
| Educational status | No formal education | 21 | 4.9 |
| | Primary (Grade1-8) | 119 | 27.9 |
| | Secondary (Grade 9–12) | 124 | 29.1 |
| | Graduate | 162 | 38 |
| Occupation | Gov't employee | 105 | 24.6 |
| | NGO | 14 | 3.3 |
| | Hired in private sector | 37 | 8.7 |
| | Merchant | 39 | 9.2 |
| | Housewife | 202 | 47.4 |
| | Daily laborers | 15 | 3.5 |
| | Other | 14 | 3.3 |
| Household's family size | <5 | 233 | 54.7 |
| | ≥5 | 193 | 45.3 |
| HH Wealth Index | Highest | 123 | 28.9 |
| | Middle | 145 | 34 |
| | Lowest | 158 | 37.1 |

The overall incidence of non-adherence was 10 per 1,000 maternal-days of IFAS use (95% CI: 8.8, 16.6). The Kaplan-Meier survival curve estimate shows the survival probabilities of the maternal IFAS use and the median time to non-adherence was 74 days (95% CI: 65.33–82.67).

The log-rank test results showed that the survival curve of IFAS time to non-adherence had statistically significantly difference by women's educational status ($\chi$2 for log-rank test = 65.63, P<0.001) (Fig 2). Based on Kaplan's Meier survival estimate, the lower the households' wealth index, the higher the risk of non -adherence of women to IFAS service and the difference was statistically significant between the groups ($\chi$2 for log-rank test = 19.14, P< 0.001). Meanwhile, time to non-adherence of IFAS had statistically significantly difference by counseling status of health institutions ($\chi$2 for log-rank test = 58.83, P<0.001).

**Table 2. Pregnant women's access to information in Hosanna town, 2021 (n = 426).**

| Access to information | Not at all | Less than once | At least once |
|---|---|---|---|
| Weekly frequency of watching TV medical advice | 146 (34.3%) | 155 (36.4%) | 125 (29.3%) |
| Weekly frequency of listening to radio | 178 (41.8%) | 117(27.5%) | 131(30.8%) |
| Weekly frequency of reading medical magazine | 226 (53.1%) | 92(21.6%) | 108(25.4%) |

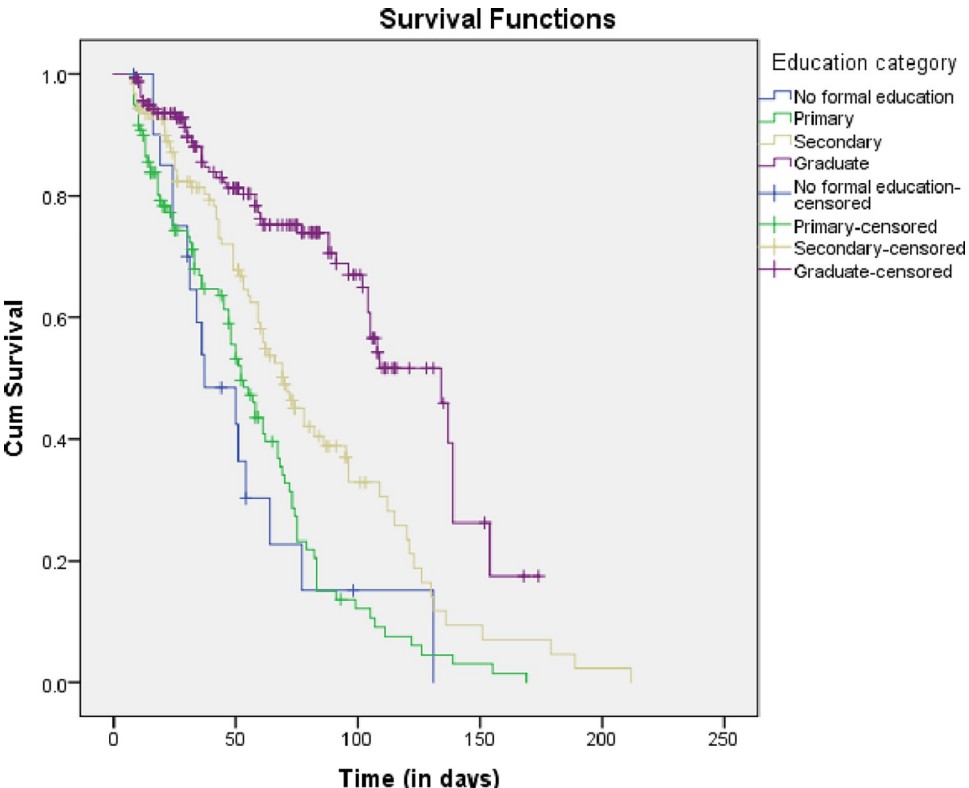

**Fig 2. Kaplan-Meier survival estimate of educational status of women taking IFAS in Hosanna town, South Ethiopia, 2021.**

### Predictors of time to non-adherence to IFAS

For factors identified as significant (p < 0.25) in the bivariable Cox regression, multivariable Cox regression analysis was performed by adjusting for confounding effects of others through stepwise backward multivariable Cox regression method. Age, education status, households' wealth index and counseling status on service delivery were found to be independent predictors of time to non-adherence of IFAS. For one year increase in age the risk of non-adherence increases by 5% (AHR = 1.05, 95% CI: 1.01–1.08). The educational status of pregnant women was significantly associated with time to non-adherence of IFAS. Pregnant women with education level having no formal education (AHR = 2.43, 95% CI 1.34–4.40), Primary education (AHR 3.00 95% CI: 2.09–4.31) and secondary, (AHR = 1.91 95% CI 1.32–2.77) are at increased risk of early non adherence when compared to their counterpart with tertiary level education. Likewise, pregnant women in middle (AHR = 1.73, 95% CI = 1.19–2.51) and lowest (AHR = 1.64, 95% CI = 1.15–2.35) wealth tertiles status are more likely to be early non-adherent when compared to pregnant women in highest wealth class. This study also witnessed that poor counseling during service delivery at health institutions increased the hazard of non-adherence to IFAS by more than 2.5 folds (AHR = 2.53, 95% CI: 1.88–3.41) (Table 3).

## Discussion

Despite its well documented benefit [7, 24, 25], this study revealed that over half of (53.1%) pregnant women in the study area were non-adherent to IFAS with the median time to non-adherence was 74 days. The early non adherence coupled with late initiation of IFAS might determine the poor pregnancy outcomes.

**Table 3. Predictors of time to non-adherence of IFAS among pregnant women in Hosanna town, South Ethiopia, 2021 (n = 426).**

| Variables | | Survival status | | Total | CHR (95% CI) | AHR (95%CI) |
|---|---|---|---|---|---|---|
| | | Event | Censored | | | |
| Age | | 226 | 200 | 426 | 1.04 (1.012–1.071)* | 1.05 (1.02–1.08) * |
| Education status | No formal education | 16 | 5 | 21 | 4.24 (2.4–7.5)** | 2.43 (1.34–4.4)* |
| | Primary (Grade 1–8) | 87 | 32 | 119 | 3.72 (2.61–5.31)** | 3.00 (2.09–4.31)** |
| | Secondary (Grade 9–12) | 75 | 49 | 124 | 2.18 (1.51–3.14)** | 1.91 (1.32–2.77)* |
| | Graduate | 48 | 114 | 162 | 1 | 1 |
| Family Size | Less than 5 | 126 | 107 | 233 | 0.81 (0.62–1.05) | |
| | Five or more | 100 | 93 | 193 | 1 | |
| Households wealth status | Highest | 50 | 73 | 123 | 1 | 1 |
| | Middle | 77 | 68 | 145 | 1.602 (1.12–2.29)* | 1.73 (1.19–2.51)* |
| | Lowest | 99 | 59 | 158 | 2.11 (1.5–2.97)** | 1.64 (1.15–2.35)* |
| Current pregnancy planned | Yes | 183 | 180 | 363 | 1 | |
| | No | 43 | 20 | 63 | 1.86 (1.33–2.6)** | |
| Reported anemia in current pregnancy | Yes | 75 | 38 | 113 | 1 | |
| | No | 151 | 162 | 313 | 1.40 (1.06–1.85)* | |
| Counseling during Service delivery | Poor | 154 | 61 | 215 | 2.91 (2.18–3.88)** | 2.53 (1.88–3.41)** |
| | Good | 72 | 139 | 211 | 1 | 1 |
| Waiting time to receive service | ≤ 30 minutes | 71 | 90 | 161 | 1 | |
| | > 30 minutes | 155 | 110 | 265 | 1.24 (0.94–1.65) | |
| Weekly Frequency of on TV medical advice | Not at all | 107 | 39 | 146 | 2.07 (1.49–2.87)** | |
| | Less than once | 65 | 90 | 155 | 1.02 (0.71–1.45) | |
| | At least once | 54 | 71 | 125 | 1 | |
| Knowledge on IFAS | Poor | 159 | 93 | 252 | 2.13 (1.59–2.85)** | |
| | Good | 67 | 107 | 174 | 1 | |

*= p-value <0.05,

** = p-value<0.01.

This study showed that for a year increase in age, there is 5% increased risk of early non adherence. This is consistent with evidences that advanced maternal age is associated with non-adherence to IFAS [3, 20, 26], indicated that women in the early reproductive ages give due attention to health care advice and as the age and the number of pregnancy increases, women become negligent; consequently become non-adherent to the supplementation during pregnancy. In our context this can be explained by the same statement that increase in age is associated with the risk of non-adherence to the supplementation. However, other studies [27–29] indicated that increase in age is positively associated with IFAS compliance. This variation could be due to socio-cultural difference of the study setup. Using tertiary educational status as a reference, there is increased hazard of early non adherence among women with no formal education, primary education and secondary education. This is in line with findings from studies [3, 19] in which educational status is associated with IFAS compliance. Additionally, this is in line with a study done in Tanzania [30], that indicated educated women are more likely than their counterparts to be knowledgeable and take advantage of utilizing the supplementation during pregnancy.

The possible explanation is that literacy status could determine understanding of information, independence, self-confidence and decisions power [31], which in turn could affect adherence to medical instructions [32].

In agreement with studies in Pakistan [3], Tanzania [15] and Ethiopia [4], this study revealed that there is increased hazard of non-adherence to IFAS among women from bottom and middle wealth tertiles compared to their counterpart women from upper wealth class. Socioeconomic status (SES) is a factor that may have an impact on adherence to health behavior. This is in line with evidence [33], suggesting that lower SES may contribute to poor adherence to health recommendations by influencing individuals' lifestyle, personal health behaviors, and access to health information. Additionally, the rationale for this might rely on the effect of cost charged for transportation for subsequent ANC visits [31]. Therefore, indirect costs might increase the hazard of non-adherence to IFAS among underprivileged wealth tertiles.

Counseling status during service delivery in health institutions was significantly associated with the hazard of non-adherence to IFAS among pregnant women. This is in line with a study in India [9] indicated that inadequate counseling is a barrier for IFAs compliance and in Tanzania [15] counseling is significantly associated with IFAS compliance. A study in Ethiopia [14] discussed that poor counseling during supplementation could adversely affect the IFAS utilization. This might be explained by the likelihood that appropriate IFAS counseling could enhance knowledge of the purpose, relevance, potential side effects, duration, and dosage of the supplement, which might subsequently have an effect on the client's adherence to the supplement.

However, in the adjusted model, women's IFAS knowledge does not show statistically significant difference of hazard in time to non-adherence. This could be due to the condition that knowledge is embedded in the educational status which showed significant difference of hazard on time to non-adherence. Moreover, the status of pregnancy plan in the current pregnancy, history of adverse fetal outcomes, history of anemia and frequency of ANC visit [4, 21] do not confer additional benefit in avoiding the hazard of time to non-adherence of IFAS among pregnant women.

## Conclusion

The median time to non-adherence was short and women became non-adherent before the recommended duration. Pregnancy plan has no evidence of better IFAS utilization. With better educational status and counseling, the median time to non-adherence would be extended to gain optimal benefit from iron folate supplementation. Educating pregnant women on the benefit of IFAS throughout the pregnancy would make a change.

### Strength and limitations

To the best of our knowledge, the current study is the first to determine time to non-adherence of IFAS. The current study revealed median time that pregnant women turning to non-adherent in the course of IFAS intake during pregnancy. Despite the significant contribution, this research has limitations. We could have not avoided the flat slop syndrome that might have inflated the censored cases. The use of one-week intake to measure the event may not reveal the reality.

### Supporting information

**S1 Data. Data collection tool.**
(DOCX)

**S2 Data. SPSS data set.**
(SAV)

## Acknowledgments

It is our sincere pleasure to acknowledge Institutional Review Board (IRB) of Hosanna Health Science College for thoroughly reviewing this work. We are gratitude to acknowledge all study participants, data collectors and supervisors for their contribution towards the accomplishment of this paper.

## Author Contributions

**Conceptualization:** Belay Bancha, Legese Petros, Habtamu Hassen, Admasu Jemal.

**Data curation:** Belay Bancha.

**Formal analysis:** Belay Bancha, Bereket Abrham Lajore, Habtamu Hassen.

**Methodology:** Belay Bancha, Legese Petros, Admasu Jemal.

**Software:** Belay Bancha, Bereket Abrham Lajore.

**Writing – original draft:** Belay Bancha, Bereket Abrham Lajore, Legese Petros, Habtamu Hassen.

**Writing – review & editing:** Belay Bancha, Bereket Abrham Lajore.

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
