## [Decision Letter · Decision Letter 0]

24 May 2022

PONE-D-22-06567Time to non-adherence of iron folic acid supplementation and associated factors among pregnant women in hosanna town, South Ethiopia: Cox-proportional hazard model.PLOS ONE

Dear Dr. Bancha,

Thank you for submitting your manuscript to PLOS ONE. After careful consideration, we feel that it has merit but does not fully meet PLOS ONE’s publication criteria as it currently stands. Therefore, we invite you to submit a revised version of the manuscript that addresses the points raised during the review process.

We look forward to receiving your revised manuscript.

Kind regards,

Linglin Xie

Academic Editor

PLOS ONE

Journal Requirements:

3. You indicated that you had ethical approval for your study. In your Methods section, please ensure you have also stated whether you obtained consent from parents or guardians of the minors included in the study or whether the research ethics committee or IRB specifically waived the need for their consent

Reviewers' comments:

Reviewer's Responses to Questions

**Comments to the Author**

1. Is the manuscript technically sound, and do the data support the conclusions?

Reviewer #1: Partly

2. Has the statistical analysis been performed appropriately and rigorously? 

Reviewer #1: Yes

3. Have the authors made all data underlying the findings in their manuscript fully available?

Reviewer #1: No

4. Is the manuscript presented in an intelligible fashion and written in standard English?

Reviewer #1: No

5. Review Comments to the Author

Reviewer #1: The manuscript needs details descriptions on certain areas, and a thorough grammatical checks. It has potential to be published if the details can be incorporated especially in the methods and results sections. I have additional comments as follows:

Title

Suggest changing title to describe clearly it is iron and folic acid. Current title is rather misleading especially the phrasing of “iron folic acid”

Abstract

- Background could highlights the supplementation nonadherence issues in association to poor pregnancy outcomes

- could be better described in terms of brief methods used/factors assessed

- use of abbreviation in conclusion when it wasn’t introduced previously anywhere in manuscript should be avoided

Introduction

- line 67-70: does this definition in reference to also Ethiopian government?

- line 73: gap of evidences was not discussed, vaguely stated – as in what was being carried out previously in reference to non-adherence and the associated factors, so the need of the study was not clearly justified

- also, tenses in general needs re checking

Methods

- suggest revising the description of setting to a more relevant information pertaining location

- population was not clearly described, especially on the inclusion including age range, trimesters

- line 88-90: not clear on which pregnant women were included in the study

- suggest changing the use of word kebele for comprehension of wider audience

- multisampling can be better visualised with use of flowcharts – suggest to include 1

- line 118-121: vaguely described the tools used – detailed explanation required

- operational definitions – arrangement of this information is out of place, with no references to which definitions were referred to

- data QA was carried out, but all the information provided were not described as part of the tools used, so this information makes no sense

- data analyses were described in details, however it was difficult to follow when tools used/factors assessed were not adequately described

Results

- line 179 suggest to remove under the study

- line 181 missing %

- readers could understand better of wealth index status if this was described in methods clearly

- figure 1 - and 2 – wrongly spelled education, labelled incorrectly in terms of numbers, and don’t see the need for including the histogram on reasons why women skipped supplementation when this could be presented in just tables

- maternal characteristics can be described under study population background

- results were not described adequately before the presentation of findings from analyses eg; descriptive of each of the factors

Discussion

- line 255- 259 were repetitions from previous sections – suggest to remove

- line 259 suggest to replace word “witnessed”

- line 261: median time was first mentioned in discussion not in results where it should be clearly presented as this would be the primary outcome?

- should include wide range of supporting literatures, and include the mechanism explanation to reason the findings observed

6. PLOS authors have the option to publish the peer review history of their article (what does this mean?). If published, this will include your full peer review and any attached files.

Reviewer #1: No

---

## [Author Response · Author response to Decision Letter 0]

8 Jul 2022

Dear Reviewers, it is our great pleasure to mention that your comments and corrections are very valuable for further progress to this paper. Following, we are mentioning some important responses and amendments as per comments and adhering to the journal guideline

• Iron and folic acid supplementation recommendation and definition also holds true in Ethiopian government

• In the manuscript we have mentioned that all pregnant women who booked for ANC one-week prior the study were included and also exclusion mentioned. 

• The Administrative structure in Ethiopia includes “kebele” for which we couldn’t find the corresponding English analogue and considering this we mentioned that it is a least administrative structure

• For tools and factors assessed we are supplementing it with the tool we have used in the study. 

• We believed that line 229–235 best fits in methods, we've relocated it to the section under the subheading of “Data Processing and Analysis”.

• We have made the data set and tools used available in this version.

• For others we did all the effort to adhere to the journal guideline and recommendations proposed by the authors. The revised version is submitted in with a track change and also without track change. Hope that this revised version will satisfy the journal standard. 

 Thanks in advance

---

## [Decision Letter · Decision Letter 1]

31 Aug 2022

PONE-D-22-06567R1Time to non-adherence to iron and folic acid supplementation and associated factors among pregnant women in Hosanna town, South Ethiopia: Cox-proportional hazard model.PLOS ONE

Dear Dr. Bancha,

Thank you for submitting your manuscript to PLOS ONE. After careful consideration, we feel that it has merit but does not fully meet PLOS ONE’s publication criteria as it currently stands. Therefore, we invite you to submit a revised version of the manuscript that addresses the points raised during the review process.

We look forward to receiving your revised manuscript.

Kind regards,

Linglin Xie

Academic Editor

PLOS ONE

Journal Requirements:

Reviewers' comments:

Reviewer's Responses to Questions

**Comments to the Author**

1. If the authors have adequately addressed your comments raised in a previous round of review and you feel that this manuscript is now acceptable for publication, you may indicate that here to bypass the “Comments to the Author” section, enter your conflict of interest statement in the “Confidential to Editor” section, and submit your "Accept" recommendation.

Reviewer #2: (No Response)

2. Is the manuscript technically sound, and do the data support the conclusions?

Reviewer #2: Yes

3. Has the statistical analysis been performed appropriately and rigorously? 

Reviewer #2: Yes

4. Have the authors made all data underlying the findings in their manuscript fully available?

Reviewer #2: Yes

5. Is the manuscript presented in an intelligible fashion and written in standard English?

Reviewer #2: Yes

6. Review Comments to the Author

Reviewer #2: - Suggestion to be consistent with using the survival time to non-adherence or just tome to non-adherence.

- Was other pregnancy related condition captured in this study such as GDM or history of GDM, hypertension etc? Just wondering whether this could have any effect on adherence.

- What is the exclusion criteria?

7. PLOS authors have the option to publish the peer review history of their article (what does this mean?). If published, this will include your full peer review and any attached files.

Reviewer #2: No

---

## [Author Response · Author response to Decision Letter 1]

5 Sep 2022

Dear editor/reviewers, 

We acknowledge all the concerns and we took a big lesson during the revision of our manuscript. Following this, here are point by point responses to the concerns. Line number that we are using in this note is based on the revised “manuscript”.

Inconsistent use of terms/phrases: We really appreciate out reviewers’ concern and in the current version we adhered to us “time to non-adherence”. 

Data collection tool: We acknowledge our respected reviewers’ concern on GDM hypertension and the likes, but our tool didn’t capture issues related to GDM and hypertension, but it has captured data related to history of adverse fetal outcomes (abortion, still birth, LBW, preterm birth), but these have not shown statistical significance association with our primary outcome both in crude and adjusted model. Data was also collected on reported anemia in the current pregnancy, which has significant association with time to non-adherence in crude analysis but not in adjusted model. We have described these variables under the subheading of “Pregnancy related conditions (line # 231-234) and discussion section (line # 330-331). 

Exclusion inclusion: Even though we didn’t state this in as separate subheading, we mentioned excusing and inclusion criteria under the subheading of “Study design and population” (line # 91-95), given that the submission guideline doesn’t recommend separate subheading to this section line.

Tabele1: we made certain corrections in table 1 (e.g. digits, bracket type…)

Citations and referencing: 

In accordance to the reviewers' concerns, we searched for more references (#30 and 33) to strengthen the discussion. We evaluated lists of the references in both our first and revised versions of the work, but we were unable to obtain a retracted reference. We learned that the reference #1 in the previous versions was not in PubMed database and we suspected the article is not from indexed publisher; hence we removed it from the list. We used Mendeley Reference Manager to handle our references while revising the text, during which we used various apparatuses (computers). We had to reinstall the reference manager because of some technical issues we encountered throughout this process. This might have caused some alterations to our referencing style. We noticed that some references lacked the URL and name of the journal. We believe that we've fixed the problems with reference and citation in this version. 

Supplemental Material: We made a minor correction in fig. 1. Hosanna Town added at the top of the figure. In previous submission, cluster of Jalonaramo was Embeded into Bobicho, and this’s corrected in the current version. Font type was also corrected to Arial. 

Thanks for consideration!!

---

## [Editor Report · Decision Letter 2]

12 Sep 2022

Time to non-adherence to iron and folic acid supplementation and associated factors among pregnant women in Hosanna town, South Ethiopia: Cox-proportional hazard model.

PONE-D-22-06567R2

Dear Dr. Bancha,

We’re pleased to inform you that your manuscript has been judged scientifically suitable for publication and will be formally accepted for publication once it meets all outstanding technical requirements.

Kind regards,

Linglin Xie

Academic Editor

PLOS ONE
---

## [Editor Report · Acceptance letter]

15 Sep 2022

PONE-D-22-06567R2 

Time to non-adherence to iron and folic acid supplementation and associated factors among pregnant women in Hosanna town, South Ethiopia: Cox-proportional hazard model. 

Dear Dr. Bancha:

I'm pleased to inform you that your manuscript has been deemed suitable for publication in PLOS ONE. Congratulations! Your manuscript is now with our production department. 

Kind regards, 

on behalf of

Dr. Linglin Xie 

Academic Editor

PLOS ONE